# Ambulatory Risk Stratification for Worsening Heart Failure in Patients with Reduced and Preserved Ejection Fraction Using Diagnostic Parameters Available in Implantable Cardiac Monitors

**DOI:** 10.3390/diagnostics14070771

**Published:** 2024-04-05

**Authors:** Shantanu Sarkar, Jodi Koehler, Neethu Vasudevan

**Affiliations:** Medtronic, Inc., Moundsview, MN 55112, USA

**Keywords:** risk stratification, implantable cardiac monitors, heart failure

## Abstract

Background: Ambulatory risk stratification for worsening heart failure (HF) using diagnostics measured by insertable cardiac monitors (ICM) may depend on the left ventricular ejection fraction (LVEF). We evaluated risk stratification performance in patients with reduced versus preserved LVEF. Methods: ICM patients with a history of HF events (HFEs) were included from the Optum^®^ de-identified Electronic Health Record dataset merged with ICM device-collected data during 2007–2021. ICM measures nighttime heart rate (NHR), heart rate variability (HRV), atrial fibrillation (AF) burden, rate during AF, and activity duration (ACT) daily. Each diagnostic was categorized into high, medium, or low risk using previously defined features. HFEs were HF-related inpatient, observation unit, or emergency department stays with IV diuresis administration. Patients were divided into two cohorts: LVEF ≤ 40% and LVEF > 40%. A marginal Cox proportional hazards model compared HFEs for different risk groups. Results: A total of 1020 ICM patients with 18,383 follow-up months and 301 months with HFEs (1.6%) were included. Monthly evaluations with a high risk were 2.3, 4.2, 5.0, and 4.5 times (*p* < 0.001 for all) more likely to have HFEs in the next 30 days compared to those with a low risk for AF, ACT, NHR, and HRV, respectively. HFE rates were higher for patients with LVEF > 40% compared to LVEF ≤ 40% (2.0% vs. 1.3%), and the relative risk between high-risk and low-risk for each diagnostic parameter was higher for patients with LVEF ≤ 40%. Conclusions: Diagnostics measured by ICM identified patients at risk for impending HFEs. Patients with preserved LVEF showed a higher absolute risk, and the relative risk between risk groups was higher in patients with reduced LVEF.

## 1. Background

Patients living with chronic heart failure (HF) have significant morbidity and mortality despite significant advancements in management [1,2]. Diagnostic assessments related to ambulatory management of HF are mostly limited to signs and symptoms and monitoring of blood pressure and weight changes [3]. Pulmonary artery pressure monitoring is widely considered to be the best option for ambulatory HF monitoring [4,5,6,7]. However, it requires an invasive procedure, patient compliance, and intensive monitoring. Implantable medical devices, such as pacemakers, implantable cardioverter defibrillators, cardiac resynchronization therapy defibrillators, and insertable cardiac monitors (ICM), provide daily measurements of several diagnostic parameters for possible evaluation of HF status in patients [8,9,10]. A method for combining HF diagnostic information to improve the ability to identify patients at risk for HF hospitalization has been previously reported for HF patients with reduced left ventricular ejection fraction (LVEF) using data collected by various kinds of cardiac implantable electronic devices (CIEDs) [11,12,13,14,15]. Recently, a similar approach of combining multiple diagnostic parameters in a Bayesian Belief Network machine learning model for identifying patients at risk for worsening HF was reported for ICM devices in HF patients with reduced and preserved ejection fraction [16]. However, the number of patients in the development and validation cohorts was very limited.

Alongside providing the ability for diagnosis and monitoring of cardiac arrhythmia [17,18,19,20,21,22,23], current ICM devices also monitor diagnostic parameters and store aggregated daily measurements longitudinally over a long period of time. These diagnostic parameters include nighttime heart rate and daytime heart rate, atrial fibrillation burden, ventricular rate during AF, heart rate variability, and activity duration. These diagnostic parameters have been available on implanted ICM devices since 2009. With the advantages of minimal invasive procedure, minimal implantation risk [24,25], remote monitoring capabilities with no requirement for patient compliance, and a more comprehensive set of cardiac diagnostic parameters, ICM-based HF monitoring can potentially serve to bridge the gap between weight scales and pulmonary artery pressure monitoring. This retrospective study validates previously reported results showing that individual diagnostic parameters available in the Reveal LINQ™ ICM can identify patients at risk for worsening HF in an ambulatory setting in a large real-world cohort of heart failure patients with a history of HF events, including patients with various NYHA class and ejection fractions. Further, this study evaluates the difference in risk stratification performance of these individual diagnostic parameters in identifying patients at increased risk of HF events in patients with reduced versus preserved LVEF.

## 2. Methods

Patients with cardiovascular diseases were included in the Optum^®^ de-identified Electronic Health Record (EHR) dataset during 2007–2021. In this cohort, we identified patients with Medtronic Reveal LINQ™ family of ICMs and having at least 6 months of EHR data prior to implant. Patients were only included if they were also hospitalized with a primary diagnosis of HF before the implant. The device-collected data were merged with the EHR data to create a de-identified database of real-world patients. Through a methodology compliant with HIPAA’s de-identification standard, a third party determined which of those patients from the Optum^®^ EHR were available in the Medtronic DiscoveryLink data warehouse, a deidentified device data warehouse containing continuous heart rhythm data from ICM devices. For patients whose data appeared in both datasets, a combined dataset was created that met HIPAA’s de-identification standard. The Optum EHR and Medtronic databases have been described previously with respect to HF diagnostics in CIEDs and HF events [15]. All patients provided consent to use their de-identified device data for research purposes when they signed up for the Medtronic CareLink™ Network. The clinical centers that allowed the use of their patient data for research purposes then consented to the storage of patient data in a de-identified Medtronic DiscoveryLink data warehouse. This retrospective analysis using de-identified data falls into the category of non-human research and is not considered a clinical study; therefore, no institutional review board approval was indicated and it is not registered on ClinicalTrials.gov.

HF events (HFEs) were used as the endpoint in the data analysis and were defined as inpatient, observation unit, or emergency stays with a primary diagnosis of HF and IV diuresis administration. Primary diagnosis of HF was ascertained based on ICD9/ICD10 codes: 428.X, 402.01, 402.11, 402.91, 404.01, 404.03, 404.11, 404.13, 404.91, 404.93, I50.X, I11.0, I13.0, and I13.2.

The diagnostic parameters—nighttime heart rate (NHR) and daytime heart rate (DHR), atrial fibrillation burden (AFB), ventricular rate during AF (VRAF), heart rate variability (HRV), and activity duration (ACT)—that are available in currently approved ICM devices, thus available in this real-world database, were evaluated in this study. The subcutaneous ICM device is mostly implanted in the 4th intercostal space left of the sternum and measures the electrocardiogram (ECG) based on two electrodes separated by 4 cm. The electrode signal is input to an ECG amplifier, which filters the signal and rectifies it before an auto-adjusting thresholding mechanism is used to sense R-waves [26]. The sensed R-waves and the RR intervals are used to determine daytime and nighttime heart rate as the average of RR intervals between 8 a.m. and 8 p.m. and midnight and 4 a.m., respectively. HRV is computed as the standard deviation of the 5 min R-R median over a 24 h period—a long-term HRV measurement. HRV is not computed over periods of time when the device detects atrial fibrillation. Atrial fibrillation is detected in ICM devices based on the incoherence of RR intervals and looking for single p-waves between two r-waves to reject inappropriate detections [27,28]. Activity duration is determined by counting the number of fluctuations in an accelerometer signal over a minute, and if the number of fluctuations is over a threshold (a nominal number of steps for a minute), then it is considered an active minute. The total number of active minutes is counted over a 24 h period to determine the daily activity duration. Subcutaneous impedance and respiration rate parameters reported in earlier work [16] were not available in these real-world devices and hence could not be evaluated. The threshold cut-offs for risk groups were previously defined with minor modifications for NHR and AFB features.

The individual parameter risk states for the currently available parameters defined using previously reported data [16] are shown in Table 1. For each diagnostic parameter, several features were computed based on the absolute threshold and the relative change threshold. For example, an absolute threshold feature for NHR is *ND(NHR_30_ ≥ 90 bpm) ≥ 10*, which computes the number of days NHR is ≥90 bpm in the last 30 days, and if that number was ≥10, then this feature criterion was true. Similarly, a relative change threshold feature example is CSFRACT_7_ ≤ 43 for the ACT feature, which computes the cumulative sum of the difference of activity measurements minus 30 min per day (a fixed reference) over the last 7 days. If this cumulative sum was ≤43, then this criterion is considered met. Multiple such feature criteria were evaluated and combined with logic to determine whether a diagnostic parameter was providing evidence for high risk. These feature criteria and thresholds were predetermined using logistic regression for each feature with respect to HF events based on development set data in earlier reports [16].

The risk stratification performance of the HF score was evaluated by simulating monthly follow-ups similar to previously reported studies [12,16], which consisted of looking at the individual parameter risk states in the last 30 days and evaluating the occurrence of clinical events in the following 30 days (Figure 1). A marginal Cox proportional hazards model was used to compare HF events for different risk groups (high, medium, and low) for each individual diagnostic parameter and estimate hazard ratios. For patients who had LVEF information available in the database, two cohorts of patients were created: one with LVEF ≤ 40% (HF patients with reduced EF–HFrEF) and one with EF > 40% (HF patients with preserved EF–HFpEF). If patients had multiple EF measurements, then the median LVEF was used. The risk stratification performance evaluation is repeated for these two cohorts of patients.

## 3. Results

A total of 1020 LINQ ICM patients with a history of HF admission prior to implant were identified from the deidentified real-world dataset. The baseline characteristics of the included patients are shown in Table 2. LVEF was available in 889 (87%) of the 1020 patients, of which 267 (30%) had EF ≤ 40%, and 622 (70%) had EF > 40%. The NYHA class was available in 296 (29%) of the 1020 patients, of which 39 (13%) were in class I, 126 (43%) were in class II, 112 (38%) were in class III, and 19 (6%) were in class IV. There were a total of 18,383 follow-up months in the dataset. A total of 301 monthly evaluations (1.6%) had an HF event in the next 30 days.

Table 3 shows the event rate comparisons between the different risk groups for individual diagnostic parameters. While the absolute event rate is lower in the different risk groups due to the lower incidence rate of HF events in this real-world patient cohort, the relative risk is very similar to previously reported results [16] for these individual diagnostic parameters. Further, the absolute risk increases if the incidence rate of HF events increases when only patients who had HF events in the year prior to the implant are included. ACT diagnostic evaluations in the high-risk group comprised 25% of all evaluations, with an event rate of 2.5% in patients in this data cohort. The activity risk status evaluations in the “high” group were 4.2 times more likely to have an HF event in the next 30 days compared to diagnostic evaluations with a low-risk status. Similarly, NHR diagnostic high-risk evaluations (26% of all evaluations) had an HF event rate of 2.9%, with high-risk evaluations being 5.0 times more likely to have an HF event in the next 30 days compared to diagnostic evaluations with a low-risk status. The HRV diagnostic evaluations considered high-risk comprised 14% of all evaluations with an event rate of 2.6% and were 4.5 times more likely to have an HF event in the next 30 days compared to low-risk diagnostic evaluations. Finally, the AF diagnostic high-risk evaluations (12% of all evaluations) had an event rate of 3.1% and were 2.3 times more likely to have an HF event in the next 30 days compared to low-risk diagnostic evaluations.

The risk stratification performance for patients with a history of HF events prior to the ICM implant and with a reduced ejection fraction (median LVEF ≤ 40%) and those for patients with preserved LVEF (median LVEF > 40%) are shown in Figure 2A,B, respectively. There were a total of 267 patients with median LVEF ≤ 40% (64 ± 13 years old, 67% male) and 622 patients with LVEF > 40% (69 ± 12 years old, 47% male). For the reduced LVEF cohort, there were a total of 4867 follow-up months, and 65 monthly evaluations (1.3%) had an HF event in the next 30 days. For the preserved LVEF cohort, there were a total of 11,057 follow-up months, with 224 monthly evaluations (2.0%) having an HF event in the next 30 days. Patients with preserved LVEF showed a higher event rate and hence a higher absolute risk for each risk group; however, the relative risk between risk groups and hence the risk stratification performance for identifying risk for worsening HF was better in patients with reduced LVEF for these diagnostic parameters.

Temporal characteristics of the ensemble average of different diagnostic parameters prior to, during, and after HF events are shown in Figure 3 for patients with HFrEF and HFpEF. There was a significant difference in changes in heart rate, with HFrEF patients exhibiting a longer-term increase in heart rate compared to a sudden increase in the case of HFpEF patients. Also, a higher AF burden was observed in HFrEF patients compared to HFpEF, with possible occurrences of new-onset persistent AF in HFrEF patients that may trigger worsening of HF symptoms. Further, HF treatment leads to a lowering of the AF burden post-HF admission in both groups.

## 4. Discussion

The results in this expanded real-world dataset show that the individual diagnostic parameters that are commercially available in an ICM device can identify when patients are at increased risk for worsening HF in a broader cohort of HF patients with a history of HF admission before the ICM implant. The diagnostic parameter feature thresholds reported previously [16] with minor pre-defined modifications for NHR and AF parameters show an increasing event rate with increasing diagnostic risk as expected, and thus the chosen feature set and threshold for risk categorization validated well in this large cohort of patients with various degrees of HF severity (NYHA class) and LVEF. The subcutaneous impedance and respiration rate features were not available in this dataset and will be validated in the future using data currently being collected in the ALLEVIATE-HF study (NCT04452149).

In this real-world cohort, patients with HFpEF had a higher overall event rate, hence higher absolute risk, whereas patients with HFrEF had a better risk stratification performance, i.e., better relative risk between high and low risk evaluations. The lower event rate in the HFrEF population can be attributed to the fact that most patients with HFrEF are being treated with effective therapies such as cardiac resynchronization therapy (CRT) or pharmacological therapies. Normally, patients receiving CRT therapy devices do not receive an ICM device; thus, the HFrEF patients in this cohort are possibly patients who refused the CRT implant or have a less severe disease. Differences in temporal characteristics were observed in the NHR and AF parameters prior to and after HF admission, which suggests the need for separate risk stratification features for HFrEF and HFpEF patients.

The diagnostic parameters investigated in this study have been shown to change prior to HF events [15] and have also been shown to be significant predictors of risk for HF events in multiple studies in patients with CIEDs [29,30,31,32,33,34,35,36,37,38,39]. Most of these studies used patients implanted with defibrillators or CRT therapy devices, which are indicated for patients with a reduced ejection fraction. Heart rate and HRV [29,30] (a surrogate of sympathetic tone) are known compensatory mechanisms that are particularly relevant for both HFrEF and HFpEF, though there might be some differences in the mechanisms involved in the two groups [40,41]. It is known that AF and rapid ventricular rate during AF are trigger mechanisms for worsening HF [32,33,34,35,36,37], with possibly different impacts on HFrEF and HFpEF. Activity duration [38,39] is considered a surrogate of functional capacity and worsening symptoms. In this study, each of these parameters showed risk stratification capabilities in an ambulatory setting in both groups of patients. To our knowledge, this is the largest cohort of a broader group of HF patients, particularly patients with HFpEF, investigated with respect to continuous ambulatory monitoring and the risk of worsening HF.

The overall goal for ambulatory diagnostic-based management of HF is to combine multiple diagnostic parameters into a single integrated dynamic risk score that is comprehensive and covers the different aspects of physiology, signs and symptoms, and trigger mechanisms to identify when patients are at increased risk of worsening HF [16]. In the setting of worsening HF, the heart becomes less efficient as a pump, be it because of reduced ability to contract effectively—systolic dysfunction [40] or reduced capacity to fill effectively—diastolic dysfunction [41]. Patho-physiologic mechanisms such as increased sympathetic tone, an increase in heart rate, increased fluid retention, and vasoconstriction set in to compensate for the less efficient circulation of blood. Diagnostic parameters such as resting heart rate (like NHR) and HRV measure some of these physiologic changes. Other diagnostic parameters, such as intra-thoracic impedance [8,9,10] in CIEDS can measure increased plasma volume, and subcutaneous impedance [16] in ICMs can measure increases in interstitial volume, thus measuring increases in fluid retention. Normally, patients with chronic HF are in a compensated state, but a trigger such as exertion, AF with rapid ventricular rate, a lung infection, or a high sodium diet can perturb this compensatory equilibrium. When compensatory mechanisms are not sufficient to handle the perturbation due to a trigger, they go into overdrive, leading to acute decompensated heart failure and the development of worsening signs and symptoms of HF such as shortness of breath, fatigue, and pulmonary and peripheral edema. These signs and symptoms can be measured using diagnostic parameters such as activity duration, one of the parameters of this study, or other parameters such as subcutaneous impedance and respiratory rate [16].

Additionally, ICMs are capable, or would be able to be in the future, of making additional measurements that are related to HF, such as PVC burden, sustained tachycardia, nighttime sleeping posture, chronotropic incompetence, R-wave morphological characteristics (amplitude, slope, and width), short-term HRV, heart sounds, oxygen saturation, temperature, systolic and diastolic intervals, etc., which can all be potentially combined to improve the performance of the risk score. Larger datasets need to be collected to further improve the algorithms, such as using deep learning neural networks, and validate the algorithms in larger independent datasets. Finally, whether a risk score-based remote ambulatory management of HF patients will lead to improved patient outcomes needs to be evaluated prospectively, as is currently being investigated in the ALLEVIATE-HF study (NCT04452149).

### Limitations

ICM devices are normally implanted for indications of unexplained syncope, cryptogenic stroke, palpitations, or AF management. HF is an incidental disease in these patients, and hence this cohort may exhibit selection bias-related differences compared to the patient cohort in previously reported data from prospective studies in primarily HF patients [16]. Additionally, this is a retrospective observational study in a real-world cohort of patients, which potentially creates bias as well. Overall, the HF event incidence rate of 1.6% in this real-world cohort is lower compared to previously reported studies. Further, due to the real-world nature of EHR data, some events may not be included in the data cohort if patients visit a hospital outside the network, leading to a possible lower event rate in higher-risk groups. Additionally, HF events cannot be adjudicated by a committee in an electronic health record-based real-world study; thus, the HF events may be “noisier” compared to events collected in previously reported prospective studies [16], leading to a higher event rate in the low-risk groups. While IV diuretics were required for HF events, incorrect ICD9/ICD10 coding may lead to the incorrect assignment of primary HF events. This may all lead to reduced hazard ratios between the risk groups.

## 5. Conclusions

A risk stratification method to identify when patients are at increased risk for worsening HF using diagnostic data collected by currently approved ICM devices was validated in an independent real-world dataset. The diagnostic parameters AF, NHR, activity duration, and HRV may be able to identify the risk of worsening HF. Relative risk between risk groups, and hence the risk stratification performance for identifying risk for worsening HF, was better in ICM patients with HFrEF.

## Figures and Tables

**Figure 1 diagnostics-14-00771-f001:**
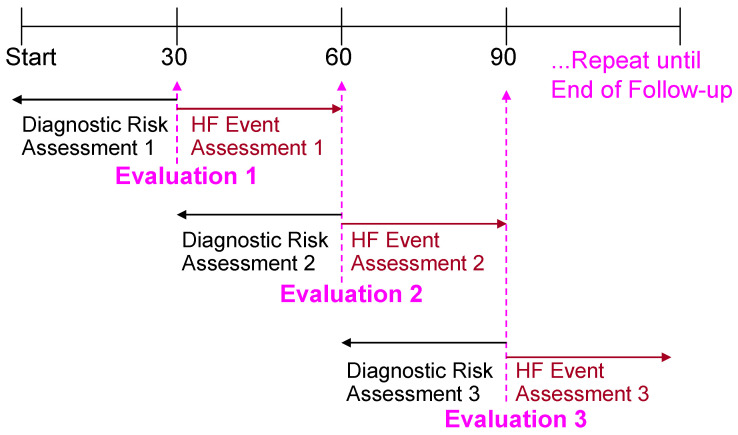
Monthly evaluation scheme for risk stratification evaluation. At every simulated monthly follow-up diagnostic evaluation is done from data in last 30 days and clinical events are evaluated in the following 30 days.

**Figure 2 diagnostics-14-00771-f002:**
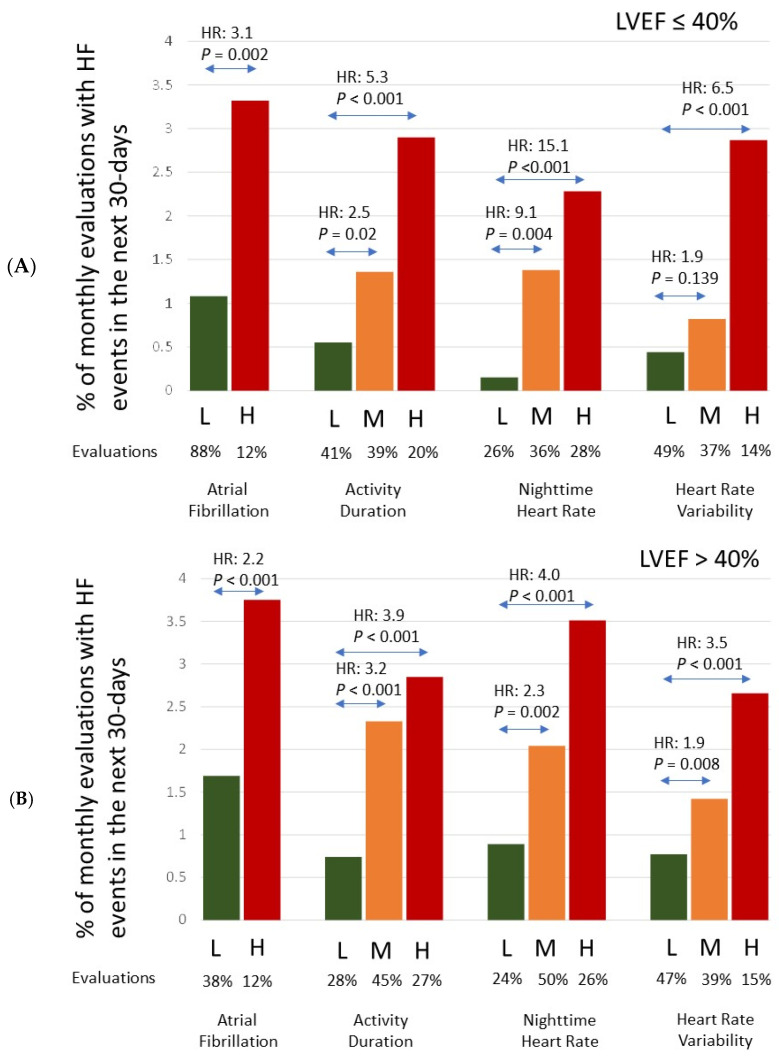
HF event rate in the different risk groups for each diagnostic parameter in patients with (**A**) LVEF ≤ 40 and (**B**) LVEF > 40. Hazard Ratios (HR) are reported for comparison of the High (H) and Medium (M) risk states to the Low (L) risk state as reference.

**Figure 3 diagnostics-14-00771-f003:**
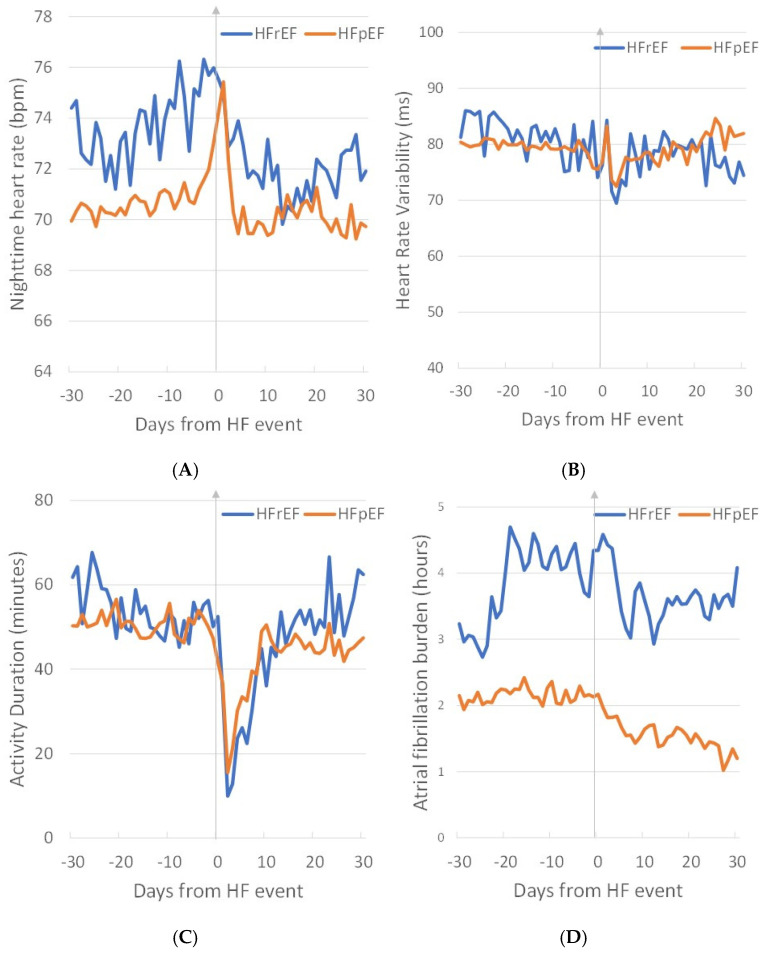
Ensemble average of diagnostic parameters (**A**) nighttime heart rate, (**B**) heart rate variability, (**C**) activity duration, and (**D**) atrial fibrillation burden in patients with HFrEF and HFpEF prior to, during, and after HF events.

**Table 1 diagnostics-14-00771-t001:** Previously defined features and their combinations to identify different levels of diagnostic evidence.

Diagnostic Evidence		Feature Set	Rationale
AF	H	[{avgAFB_7_ > 12.5 h OR avgAFB_30_ > 12 h OR avgAFB_7—_avgAFB_30_ > 0.6 h OR ND(AFB_30_ > 6 h) ≥ 1 OR ND(AFB_7_ > 6 h) ≥ 1} AND ND(AFB_30_ > 23 h) < 30]	Presence of paroxysmal AF
		OR max2minAFB_7_ > 6.5 h OR max2minAFB_30_ > 7 h	Change in AF burden
		OR ND(AFB_30_ > 6 h AND VRAF_30_ ≥ 90 bpm) ≥ 1 OR ND(AFB_30_ > 23 h AND VRAF_30_ ≥ 90 bpm) ≥ 15	Poor rate control
		OR maxVRAF(AFB_30_ > 6 h) ≥ 80 bpm OR maxVRAF(AFB_7_ > 6 h) ≥ 70 bpm	Higher rates during AF
	L	Not ‘H’	
NHR	H	maxNHR_30_ > 95 bpm OR minNHR_30_ < 40 bpm OR minNHR_30_ ≥ 80 bpm OR ND(NHR_30_ ≥ 90 bpm) ≥ 10	High resting heart rate
		OR {avgNHR_7—_avgNHR_30_} ≥ 8 bpm	Increasing resting HR
		OR max2avgNHR_30_ ≥ 33%	Change in resting HR
	M	Avg(DHR-NHR)_30_ < 9 bpm} AND Not ‘H’	Similar daytime and nighttime HR
	L	{Not ‘H’ OR ‘M’}	
HRV	H	ND(HRV_30_ ≤ 60 ms) ≥ 25 OR minHRV_7_ < 35 ms OR CSFRHRV_30_ < −12	Very low HRV or high sympathetic tone
		OR max2avgHRV_30_ ≥ 85%	Change in HRV
	M	{ND(HRV_7_ ≤ 60 ms) ≥2 OR ND(HRV_30_ ≤ 60 ms) ≥ 6 OR minHRV_30_ < 55 ms OR avgHRV_30_ < 65 ms OR avgHRV_7_ < 75 ms OR CSFRHRV_7_ < −2	Lower HRV or higher sympathetic tone
		OR max2avgHRV_30_ ≥ 65% }	Change in HRV
		AND Not ‘H’	
	L	{Not ‘H’ OR ‘M’}	
ACT	H	ND(ACT_7_ ≤ 30 min) ≥ 7 OR ND(ACT_30_ ≤ 30 min) ≥ 27 OR avgACT_7_ < 10 min OR CSFRACT_7_ < −43	Very low ACT or low functional capacity
	M	{ND(ACT_30_ ≤ 30 min) ≥ 11 OR avgACT_30_ < 30 min OR CSFRACT_30_ < −3	Lower ACT or lower functional capacity
		OR max2avgACT_30_ ≥ 150% }	Change in ACT
		AND Not ‘H’	
	L	Not ‘H’ OR ‘M’	

Max: maximum; min: minimum; avg: average; ND: number of days; feature metric subscript: look back window size; CSFR: cumulative sum fixed reference; AFB: atrial fibrillation burden; VRAF: ventricular rate during AF; NHR: nighttime heart rate; DHR: daytime heart rate; HRV: heart rate variability; and ACT: daily activity duration.

**Table 2 diagnostics-14-00771-t002:** Baseline characteristics of included patients.

	All Patients with a History of HF Events	Patients with a History of HF Events and A Median LVEF ≤ 40	Patients with a History of HF Events and A Median LVEF > 40
Number of patients	1020	267	622
Mean age (SD)	68 (13)	64 (13)	69 (12)
Male gender	535 (52%)	178 (67%)	294 (47%)
Hypertension	967 (95%)	249 (93%)	599 (96%)
Diabetes	571 (56%)	133 (50%)	362 (58%)
CAD	766 (75%)	212 (79%)	464 (75%)
MI	403 (40%)	105 (39%)	264 (42%)
Vascular disease	312 (31%)	78 (29%)	205 (33%)
Atrial fibrillation	586 (57%)	158 (59%)	358 (58%)
Renal dysfunction	539 (53%)	130 (49%)	352 (57%)
Stroke/TIA	525 (51%)	117 (44%)	353 (57%)
Medications			
ACE-I/ARB	855 (84%)	245 (92%)	512 (82%)
Beta-Blockers	797 (78%)	202 (76%)	510 (82%)
Diuretics	915 (90%)	250 (94%)	557 (90%)
Spironolactone	532 (52%)	142 (53%)	336 (54%)
Sacubitril/valsartan	26 (3%)	24 (9%)	2 (0.3%)
Vasodilator/Nitrate	256 (25%)	104 (39%)	130 (21%)
AAD Class I	72 (7%)	11 (4%)	52 (8%)
AAD Class III/IV	382 (37%)	123 (46%)	228 (37%)
Anticoagulation	547 (54%)	156 (58%)	332 (53%)
ICM Reason for monitoring			
AF ablation monitoring	52 (5%)	19 (7%)	23 (4%)
AF management	172 (17%)	55 (21%)	99 (16%)
Cryptogenic stroke	240 (24%)	59 (22%)	160 (26%)
Palpitations	47 (5%)	10 (4%)	29 (5%)
Suspected AF	71 (7%)	21 (8%)	46 (7%)
Syncope	388 (38%)	83 (31%)	240 (39%)
Ventricular tachycardia	27 (3%)	13 (5%)	13 (2%)
Other/unknown	23 (2%)	7 (3%)	12 (2%)

**Table 3 diagnostics-14-00771-t003:** HF event rate comparison between different risk groups for individual diagnostic parameters in ICM patients with a history of HF events prior to the implant.

Diagnostic Parameter/Risk State	Number of Evaluations (%)	Number of HF Events (% of Evals)	Hazard Ratio (95% CI)	*p*-Value
AF				<0.001
Low	15,716 (88%)	216 (1.37%)	Reference	
High	2135 (12%)	67 (3.14%)	2.30 (1.59–3.33)	
Activity				<0.001
Low	5618 (31%)	34 (0.61%)	Reference	
Medium	8124 (44%)	150 (1.85%)	3.07 (2.05–4.58)	<0.001
High	4641 (25%)	117 (2.52%)	4.20 (2.68–6.58)	<0.001
NHR				<0.001
Low	4746 (25%)	28 (0.59%)	Reference	
Medium	9321 (49%)	151 (1.62%)	2.76 (1.70–4.46)	<0.001
High	5060 (26%)	148 (2.92%)	5.00 (3.13–8.00)	<0.001
HRV				<0.001
Low	8406 (48%)	48 (0.57%)	Reference	
Medium	6545 (37%)	75 (1.15%)	2.01 (1.35–2.99)	0.001
High	2532 (14%)	65 (2.57%)	4.53 (2.83–7.25)	<0.001

## Data Availability

Because of contractual arrangements between Optum^®^ and Medtronic Inc., the data and study materials cannot be made available to other researchers for purposes of reproducing the results or replicating the procedure.

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
