# Peer review of "Ambulatory Risk Stratification for Worsening Heart Failure in Patients with Reduced and Preserved Ejection Fraction Using Diagnostic Parameters Available in Implantable Cardiac Monitors"

_diagnostics, 2024, doi:10.3390/diagnostics14070771_

Round 1

Reviewer 1 Report

Comments and Suggestions for Authors

Allways is necesary a way to improve our capability to idéntify the risk of patients with HFrEF r 

Author Response

Authors would like to thank the reviewer for their review and considering this manuscript acceptable for publication.

No comments to address, hence no modifications made to manuscript.

Reviewer 2 Report

Comments and Suggestions for Authors

    In this intriguing paper, the authors investigated 1020 heart failure patients previously hospitalized, aiming to assess the effectiveness of nighttime heart rate (NHR), heart rate variability (HRV), atrial fibrillation (AF) burden, rate during AF, and activity duration (ACT) daily in predicting HF events. 

HF events were defined as inpatient stays with a primary diagnosis of HF, or observation unit or emergency stays with a primary diagnosis of HF and administration of IV diuresis.The diagnosis of HF relied on the ICD code of the hospitalization.

The present study provides important data and demonstrates findings consistent with other CIED studies. This study is unique in its focus on patients with loop recorders.

Methods and results are clearly presented, and the points are well discussed in the discussion section.

Major:

The limitations section is well written, but as a major point, I would emphasize that this is an observational study.

The investigators seemingly relied on ICD-9 codes to define their events. This is the major weakness of the study, as incorrect coding by the primary physician could negatively impact the results. Cardiac events were apparently not reviewed by the investigators.

The conclusion is well written, but given the observational nature of the study and its high potential for bias, I would conclude that the parameters (AF, NHR, Activity duration, and HRV) "may be able to identify…". 

No need to mention that HFpEF patients had more events, as this is a selection bias, as mentioned in the discussion.

Minor:

Table 01 is one of the most important aspects of the paper, but the information is not clearly displayed. Please reformulate the layout.

Author Response

Authors would like to thank the reviewer for their review and considering this manuscript acceptable for publication with minor revision.

The reviewer comments are addressed as follows:

Major:

The limitations section is well written, but as a major point, I would emphasize that this is an observational study.

Authors response: The following statement was included in the limitations section - "Additionally, this is a retrospective observational study in a real-world cohort of patients which potentially creates bias as well." 

The investigators seemingly relied on ICD-9 codes to define their events. This is the major weakness of the study, as incorrect coding by the primary physician could negatively impact the results. Cardiac events were apparently not reviewed by the investigators.

Authors response: The following statement was included in the limitations section - "While IV diuretics were required for HF events, however incorrect ICD9/ICD10 coding may lead to incorrect assignment of primary HF events."

The conclusion is well written, but given the observational nature of the study and its high potential for bias, I would conclude that the parameters (AF, NHR, Activity duration, and HRV) "may be able to identify…". 

Authors response: The following statement in conclusion has been modified as per recommendation - The diagnostic parameters, AF, NHR, Activity duration, and HRV may be able to identify risk for worsening HF.

No need to mention that HFpEF patients had more events, as this is a selection bias, as mentioned in the discussion.

Authors response: The statement has been removed from conclusion.

Minor:

Table 01 is one of the most important aspects of the paper, but the information is not clearly displayed. Please reformulate the layout.

Authors response: Table 1 has been updated to make it more readable.

Reviewer 3 Report

Comments and Suggestions for Authors

First of all, I congratulate the authors on the work they have put into preparing the manuscript. I recommend its adoption in its present form. 

Author Response

(The authors gave the same response as above.)

Reviewer 4 Report

Comments and Suggestions for Authors

Ambulatory risk stratification for worsening heart failure (HF) using insertable cardiac monitors is crucial for the prevention of heart failure decompensation and progression, so the manuscript can add some information to our knowledge in this topic.

Author Response

(The authors gave the same response as above.)
